# Correlation of Presynaptic and Postsynaptic Proteins with Pathology in Alzheimer’s Disease

**DOI:** 10.3390/ijms25063130

**Published:** 2024-03-08

**Authors:** Geidy E. Serrano, Jessica Walker, Courtney Nelson, Michael Glass, Richard Arce, Anthony Intorcia, Madison P. Cline, Natalie Nabaty, Amanda Acuña, Ashton Huppert Steed, Lucia I. Sue, Christine Belden, Parichita Choudhury, Eric Reiman, Alireza Atri, Thomas G. Beach

**Affiliations:** 1Civin Laboratory for Neuropathology, Banner Sun Health Research Institute, Sun City, AZ 85351, USA; jessica.walker2@bannerhealth.com (J.W.); richard.arce@bannerhealth.com (R.A.); anthony.intorcia@bannerhealth.com (A.I.); madison.cline@bannerhealth.com (M.P.C.); nlnabaty@arizona.edu (N.N.); amacuna2@asu.edu (A.A.); ahuppert.com@arizona.edu (A.H.S.);; 2Cleo Roberts Center, Banner Sun Health Research Institute, Sun City, AZ 85351, USAparichita.choudhury@bannerhealth.com (P.C.);; 3The Banner Alzheimer’s Institute, Phoenix, AZ 85006, USA

**Keywords:** aging, SNAP25, PSD95, neurofibrillary tangles, amyloid plaques, Braak stage, cerebral cortex, positron emission tomography (PET) imaging

## Abstract

Synaptic transmission is essential for nervous system function and the loss of synapses is a known major contributor to dementia. Alzheimer’s disease dementia (ADD) is characterized by synaptic loss in the mesial temporal lobe and cerebral neocortex, both of which are brain areas associated with memory and cognition. The association of synaptic loss and ADD was established in the late 1980s, and it has been estimated that 30–50% of neocortical synaptic protein is lost in ADD, but there has not yet been a quantitative profiling of different synaptic proteins in different brain regions in ADD from the same individuals. Very recently, positron emission tomography (PET) imaging of synapses is being developed, accelerating the focus on the role of synaptic loss in ADD and other conditions. In this study, we quantified the densities of two synaptic proteins, the presynaptic protein Synaptosome Associated Protein 25 (SNAP25) and the postsynaptic protein postsynaptic density protein 95 (PSD95) in the human brain, using enzyme-linked immunosorbent assays (ELISA). Protein was extracted from the cingulate gyrus, hippocampus, frontal, primary visual, and entorhinal cortex from cognitively unimpaired controls, subjects with mild cognitive impairment (MCI), and subjects with dementia that have different levels of Alzheimer’s pathology. SNAP25 is significantly reduced in ADD when compared to controls in the frontal cortex, visual cortex, and cingulate, while the hippocampus showed a smaller, non-significant reduction, and entorhinal cortex concentrations were not different. In contrast, all brain areas showed lower PSD95 concentrations in ADD when compared to controls without dementia, although in the hippocampus, this failed to reach significance. Interestingly, cognitively unimpaired cases with high levels of AD pathology had higher levels of both synaptic proteins in all brain regions. SNAP25 and PSD95 concentrations significantly correlated with densities of neurofibrillary tangles, amyloid plaques, and Mini Mental State Examination (MMSE) scores. Our results suggest that synaptic transmission is affected by ADD in multiple brain regions. The differences were less marked in the entorhinal cortex and the hippocampus, most likely due to a ceiling effect imposed by the very early development of neurofibrillary tangles in older people in these brain regions.

## 1. Introduction

Neurodegenerative diseases such as Alzheimer’s disease dementia (ADD) are characterized by brain weight loss, probably due to progressive degeneration and the death of nerve cells, which causes the loss of brain matter [1,2,3]. This includes brain matter loss in areas that are associated with cognition and, thus, is a potentially useful biomarker. While brain weight loss could be explained by neuronal loss, axonal loss, or synapse loss, previous stereological studies have failed to show any significant neocortical neuronal loss in ADD [4,5,6,7,8,9]. Therefore, the greater brain weight loss that we and others have observed in ADD [10] is most likely due to axonal loss and/or synaptic loss. Synaptic integrity is affected in multiple neurodegenerative diseases, and highly correlates with cognitive decline in both human and animal models. A common hypothesis has proposed that the progression of ADD is accompanied by synapse loss, due to the accumulation of pathologic hyperphosphorylated tau and amyloid-β [11,12,13]. Beyond actual loss, some have suggested that synaptic dysfunction might precede late-stage features of many neurological conditions such as ADD [11,14,15,16,17,18,19]. The correlation between synaptic loss and Alzheimer’s disease dementia (ADD) was established in the late 1980s, using electron microscopy (EM) techniques [3]. These methods are precise but are limited by the laborious tissue processing required and by their practical restriction to extremely small tissue samples. In the 1990s, immunochemical quantification became possible and confirmed 30–50% neocortical synaptic protein losses in ADD [20]. Very recently, PET imaging of synapses is being developed, accelerating the focus on the role of synaptic loss in ADD and other conditions [6,21,22]. However, there has not yet been an extensive profiling of different synaptic proteins in different brain regions in ADD. We know that brain functions are dependent on synaptic function, which is regulated by diverse synaptic proteins that are distributed in various subcellular compartments of the synapse [20]. SNAP-25 is a presynaptic component of the Soluble N-ethylmaleimide-sensitive factor activating protein receptor (SNARE) complex, which is central to synaptic vesicle exocytosis processes at the presynaptic terminal and regulates postsynaptic receptor trafficking, spine morphogenesis, and plasticity, and reductions in SNAP-25 have been linked to intellectual impairments, psychiatric disorders, and ADD [23,24,25]. Likewise, postsynaptic protein PSD95 disruption is associated with cognitive and learning deficits, because it is also an essential component involved in the modulation of synaptic strength and plasticity [26]. In this study, we have carried out immunochemical assays to estimate the relative expression of presynaptic protein synaptosome-associated protein 25 (SNAP25) and postsynaptic density protein 95 (PSD95) in different brain regions in cognitively unimpaired subjects (CU), subjects with ADD, and subjects with moderate levels of AD pathology that were either cognitively unimpaired (CU) or had mild cognitive impairment (MCI).

## 2. Results

Table 1 shows the general clinical and pathological characteristics of the study subjects. The group age means differed significantly (*p* < 0.0001). The youngest group (ADD) had a mean age of 82, while the oldest group (CU-HP) mean age was 92. The mean ages for the CU-LP and MCI groups were 83 and 90 years, respectively. As expected from group definitions, AD pathology summary scores from combined frontal, temporal, parietal, hippocampal CA1, and entorhinal regions, were significantly different between groups, with the CU-LP group having the lowest densities of both total plaques and tangles, ADD having the highest densities, and MCI and CU-HP showing almost identical densities (Table 1 and Figure 1). MMSE scores were significantly lower in ADD compared to all other groups, while MCI group had lower scores than the CU groups, but these did not meet statistical significance (Table 1). PMI and sex distribution were not significantly different between groups, and brain weights were only significantly lower in ADD. 

Presynaptic protein SNAP25 expression was significantly reduced in ADD subjects’ frontal cortex, primary visual cortex Brodmann area 17, and cingulate, to 44%, 57%, and 38% of the CU-LP subjects’ levels, while the hippocampus showed a 62% non-significant reduction. The entorhinal cortex showed a similar protein expression in most groups except in CU-HP, which showed higher concentrations of SNAP25 in all brain regions. However, these differences did not reach statistical significance, while SNAP25 concentrations in MCI were very similar to CU-LP (Figure 2). PSD95 protein levels were significantly reduced in almost all brain regions of the ADD group, to 46%, 48%, 56%, and 47%, in the frontal, cingulate, visual, and entorhinal areas, respectively, when compared to CU-LP; the hippocampus showed a 46% reduction but did not reach the significance level. The expression of PSD95 in the MCI and CU-HP groups was greater than in the CU-LP group, but this did not reach the significance level, except for the cingulate and the visual cortex, which also presented significant increases in PSD95 in the CU-HP group when compared to the CU-LP group, and the hippocampus, which showed similar PSD95 protein concentrations in the MCI group when compared to the CU-LP group. In addition, we observed a greater synaptic protein reduction, for both SNAP25 and PSD95, in females with ADD than in males, relative to CU. These differences reached significance in the cingulate and frontal and visual cortices, while no sex differences were observed in the entorhinal cortex or the hippocampus (Figure 2). 

Univariable correlations were significant, for both SNAP25 and PSD95, with plaque and NFT densities as well as with cognition, specifically MMSE scores, and age in most brain regions (Table 2), but not with brain weight. Logistic regression models confirmed that the MMSE scores, total plaque scores, and total NFT scores correlated significantly with cortical synaptic protein concentrations, even after correcting for sex and age (Figure 3). 

## 3. Discussion

It has been accepted for decades that synapses are lost and dysregulated in ADD [5,11,14,15,16,18]; yet, to our knowledge, this is the first time that both pre- and post-synaptic densities, from multiple areas in the brain, have been immunochemically evaluated in a large set of pathologically confirmed cases. SNAP25 is a component of the SNARE complex, which is central to synaptic vesicle exocytosis, fusion, and eventually neurotransmitter release. It is suggested that chronic reductions in SNAP25 levels are associated with intellectual impairment, psychiatric disorders, and ADD [23,24,25]. In addition, some studies suggest that a reduction in this protein may impair the structure and/or function of postsynaptic proteins, such as PSD95. Postsynaptic compartments are very complex, as they constitute multiple proteins that are constantly being modulated by external and internal signals which make these structures not only susceptible to pathological changes, but also good therapeutic targets. PSD95 is suggested to be one of the most abundant postsynaptic proteins; it is located at excitatory synapses, an essential component involved in the modulation of synaptic strength and plasticity by regulating glutamatergic receptor trafficking, which is reportedly impaired in ADD [27]. Therefore, multiple studies suggest that PSD95 disruption is associated with cognitive and learning deficits. Reduced expression has been also observed in the brains of subjects who die with ADD and in AD animal models [25,26,28,29]. Even though our analysis cannot differentiate between synaptic loss vs. synaptic protein expression loss, our results support the previous general understanding of synaptic dysregulation in ADD. Specifically, our study shows that lower synaptic protein concentrations are associated with lower MMSE scores [6]. We observed considerable synaptic protein concentration variability within each group. We avoided including cases with other comorbidities, but the observed variability could be attributed to multiple factors, including sex differences and variable levels of AD pathology. In addition, we observed a possible ceiling effect on synaptic protein reduction in the entorhinal cortex and the hippocampus, areas affected by ADD early in the disease course. Total densities of plaques and tangles also correlated with the concentrations of synaptic protein, but even though we found statistically significant correlations, the correlations were not as strong as expected, suggesting other underlying mechanisms that should be further explored. Similar results were recently reported by Plachez et al., in 2023, where they showed that synaptic modulation in an AD mouse model was observed in primary and association cortices, independently of amyloid accumulation [30]. However, is important to emphasize that, in the current study, the largest synaptic protein group differences were seen in brain regions that are affected by NFT at later stages of AD.

The entorhinal area and hippocampus are major interfaces with the neocortex, making these areas crucial for memory formation and consolidation. The between-group equivalence of synaptic density in these regions suggests that synaptic loss in these regions might start during normal aging [2,3]. Therefore, we conclude that preventing or reversing synaptic loss might best be accomplished long before the first signs of cognitive decline. Our study also suggests important sex differences that we would like to explore in future studies, which would include a larger number of subjects per sex in each group. Our data showed that ADD females experience a proportionately greater loss of synapses than males [31,32,33]. Studying such differences will not only help us understand better the mechanisms through which synapses might be lost in ADD and aging, but will also be an important consideration in future treatment trails [34]. 

Another puzzling result from this study is the apparent upregulation of synaptic proteins in multiple brain regions in the MCI and CU-HP groups, suggestive of a compensatory effect at early stages of the disease. This phenomenon has been previously observed by others and could explain the resilience of a group of individuals that potentially could “tolerate” greater amounts of pathology, resulting in a delay of disease progression. However, some studies hypothesize that the upregulation of these synaptic proteins triggers detrimental events in the synaptic terminals that result in actual synaptic loss [18,34,35,36]. Screening a larger number of pathological samples or in vivo studies using currently available PET ligands might be able to answer this question; if the latter is true, we should observe higher synaptic protein levels in most CU or MCI cases with high AD pathology, not only in a subset. 

Future studies that could allow us to visually quantify synapses in multiple brain regions of the same individuals are still needed, to determine whether our results are due to the loss of synapses or the loss of protein concentrations per synapse. This will also allow us to understand better the increased synaptic protein expression in the CU-HP group. Are these subjects expressing more synaptic protein or do they have more synapses? It will be important to understand if these subjects’ cognitive abilities are still preserved, even with moderate levels of AD pathology, because they had greater pre-existing synaptic densities or synaptic protein expression, or whether these were developed as a compensatory response to progressive AD pathology. If the latter is true, this suggests that therapeutic agents might be able to be developed to initiate these responses.

## 4. Materials and Methods

### 4.1. Subject Selection and Description

Subjects were all volunteers in the Arizona Study of Aging and Neurodegenerative Disorders (AZSAND), a longitudinal clinicopathological study of normal aging, cognition, and movement in the elderly since 1996 in Sun City, Arizona [37,38]. Autopsies were performed by the Banner Sun Health Research Institute Brain and Body Donation Program (BBDP) “www.brainandbodydonationprogram.org (accessed on 1 March 2024)”. All subjects signed Institutional Review Board-approved informed consent forms allowing both clinical assessments during life and several options for brain and/or bodily organ donation after death. Most subjects were clinically characterized with annual standardized test batteries consisting of general neurological, cognitive, and movement disorders components, including the Mini Mental State Examination (MMSE) [37,38]. Subjects for the current study (Table 1; *n* = 101) were chosen by searching the BBDP database for cases that had less than six hours postmortem interval (PMI), and either a clinicopathological diagnosis of ADD (*n* = 35), mild cognitive impairment (MCI; *n* = 18), or were cognitively unimpaired (CU), defined as those lacking dementia and subdivided into two groups, with either low (CU-LP; *n* = 33) or moderate to high levels of AD pathology (CU-HP; *n* = 15). AD pathology was assessed according to the National Institute on Aging–Alzheimer’s Association guidelines for the neuropathologic assessment of Alzheimer’s disease (NIA-AA) [39]. Brain weights were determined at autopsy, after the removal of 10–30 cc of ventricular cerebrospinal fluid but prior to fixation. The complete neuropathological examination was performed using standard AZSAND methods [37,38]. Thick (40–80 μm), large-format (up to half of each cerebral hemisphere) sections of frozen, cryoprotected fixed brain were taken using a sliding freezing microtome, while standard-sized paraffin-embedded blocks were sectioned (5–6 μm) using a rotary microtome. Microscopic observations included assessment of frontal, parietal, temporal, and occipital lobes, all major diencephalic nuclei, and major subdivisions of the brainstem, cerebellum, and spinal cord (limited to cervical cord for brain-only autopsies). Both slide sets were stained with hematoxylin and eosin and the large-format set was also stained for senile plaques, neurofibrillary changes, and other neuronal and glial tauopathies using thioflavin S, Gallyas, and Campbell–Switzer methods [37,38,40,41,42]. In all cases, an additional set of paraffin sections was immunohistochemically stained for phosphorylated α-synuclein (p-syn), while staining for phosphorylated TDP-43 (p-TDP43) was done for a subset of subjects [43,44,45]. Amyloid plaque (both diffuse and neuritic), and neurofibrillary tangle (NFT) densities were graded blindly as none, sparse, moderate, or frequent, using the Consortium to Establish a Registry for Alzheimer’s Disease (CERAD) templates [46]. All scores were converted to a zero–three scale for statistical purposes. Regions scored included cortical gray matter from frontal, temporal, parietal, hippocampal CA1, and entorhinal regions, with a maximum summary score of 15. The summation of all scores from all areas were used for all statistical correlations presented in this article. Neurofibrillary degeneration was staged on the thick frozen sections by the original method of Braak [40,42], and neuropathological ADD diagnoses were made when neuritic plaque densities and Braak stage met “intermediate” or “high” criteria, according to NIA-AA criteria [27,39,47,48]. Non-ADD conditions were diagnosed using standard clinicopathological criteria, with international consensus criteria for those disorders where these were available, but cases with major comorbidities were excluded from this study. 

### 4.2. Synaptic Densities

Frozen samples consisted of 30 cryostat sections (approximately equivalent to 100 mg of tissue) from frontal cortex, cingulate, primary visual cortex Brodmann area 17, hippocampus, and entorhinal cortex. Tissue was homogenized for protein extraction in 1 mL of RIPA buffer plus protein inhibitor cocktail (PIC) using an OmniTH tissue grinder. Homogenates were then centrifuged at 40,000× *g* for 30 min at 4 °C and the supernatant was collected and stored at −80 °C. Total protein was quantified using Micro BCA Protein Assay (23235, Thermo Fisher Scientific, Carlsbad, CA, USA). Laboratory-developed sandwich enzyme-linked immunosorbent assays (ELISA) were used to measure the concentration of presynaptic protein, SNAP25, and post-synaptic protein, PSD95. The protocol is a modification from Gottschall et al., 2010 [4]. For the SNAP25 assay, mouse monoclonal anti-SNAP25, 1:200 (clone SP14, MAB331, Millipore, Burlington, MA, USA) was used as the capture antibody and polyclonal rabbit anti-SNAP25, 1:1000 (IgG fraction, S9684, Sigma, St. Louis, MO, USA) was used as the detection antibody. For PSD95, mouse anti-PSD95 antibody at 1:100 (clone 7E3-1B8, MAB1598, Millipore, Burlington, MA, USA) was the capture antibody, while rabbit anti-PSD95, 1:400 (ab18258, Abcam, Boston, MA, USA) was used as the detection antibody. Absorbance was measured at 450 nm on a Bio-Rad iMark absorbance microplate reader (Bio-Rad, Hercules, CA, USA). Standard curves were created by dilution of mixtures of cortical brain derived from two male and female control subjects. 

Briefly, capture antibody was incubated in the ELISA plates overnight at room temperature. Then, the plates were washed with buffer B (10 mM phosphate-buffered saline, pH 7.5, 0.05% Tween 20), and all wells were blocked with blocking/dilution buffer (PBS 0.05% Tween 20 +1% BSA + 50 mM glycine + PIC) for one hour with shaking. Standard curve and study protein samples were diluted in blocking/dilution buffer and then incubated in the coated plates for two hours at room temperature with shaking. After removing the samples and washing again, the plates were incubated with the detection antibody for an additional 2 h at room temperature with shaking and then washing in buffer B. Developing antibody consisted of (horseradish peroxidase conjugated) AffiniPure goat anti-rabbit immunoglobulin G (IgG) (1:10,000) (111-035-144, Jackson Laboratories, West Grove, PA, USA) diluted in blocking/dilution buffer and added as 100 µL. The plates were shaken at room temperature for 45 min. Wells were washed five times with buffer B and 100 µL of tetramethylbenzidine substrate (T8665, Sigma, St. Louis, MO, USA) was added; the plate incubated in the dark until the zero-substrate blank became a faint pale blue (usually 20 to 30 min depending on the assay). To stop development, 50 µL of 1 M H_2_SO_4_ was added, and the absorbance measured immediately at 450 nm on a Bio-Rad iMark absorbance microplate reader.

### 4.3. Statistical Methods

Univariate analyses were used as an initial screen to indicate which variables might have significant relationships with diagnostic groups, synaptic protein concentrations and/or MMSE scores. For comparing group measures, the Mann–Whitney U-Test, one way analysis of variance (ANOVA), Kruskal–Wallis ANOVA, followed by Dunnett’s multiple comparison against CU-LP, and contrast analyses were used as appropriate. The chi-squared test was used to compare proportions and Spearman’s method was used to test univariate correlations. Variables that were significantly affected on this initial screen were included in multivariable logistic regression models. 

## Figures and Tables

**Figure 1 ijms-25-03130-f001:**
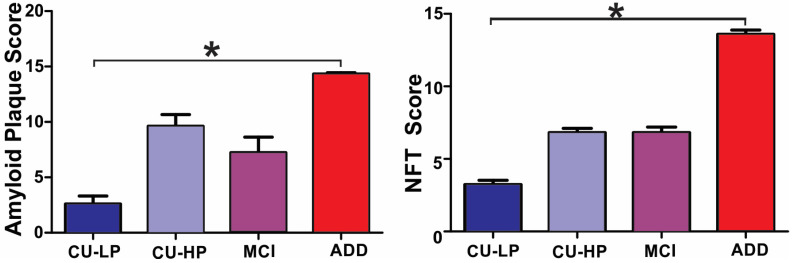
Semiquantitative scores for AD pathology. Total amyloid plaque and neurofibrillary tangle (NFT) densities were graded blindly, as described in Section 4. The highest densities of both plaques and tangles were seen in the ADD group (* ANOVA *p* < 0.05).

**Figure 2 ijms-25-03130-f002:**
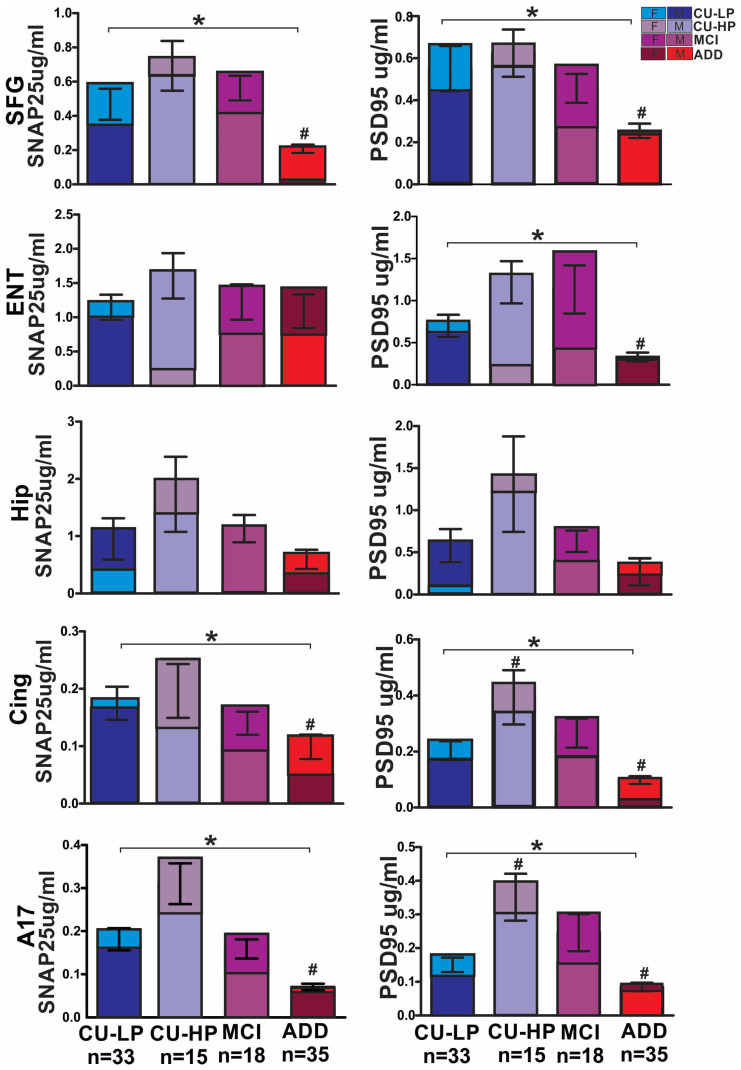
Protein expression of presynaptic protein SNAP25 and the postsynaptic protein PSD95 in the human brain, measured by ELISA. SNAP25 was significantly reduced in ADD when compared to CU-LP controls in frontal and visual cortices and the cingulate, while the hippocampus showed a non-statistically significant reduction and the entorhinal cortex showed similar protein concentrations across most groups. SNAP25 concentrations in MCI were very similar to CU-LP, while CU-HP showed a non-statistically significant increase. PSD95 concentrations were significantly lower in ADD when compared to CU-LP, in all brain regions except the hippocampus. In addition, PSD95 concentrations were significantly higher in CU-HP when compared to CU-LP in the cingulate and the visual cortex, while the hippocampus and the entorhinal cortex only showed a non-statistically significant increase. The frontal cortex showed similar PSD95 protein concentrations to those detected in CU-LP. The MCI group showed a similar non-statistically significant increase in most regions except in the hippocampus, where we observed similar protein concentrations to CU-LP. (* ANOVA *p* < 0.05; # Dunnett’s multiple comparison test against CU-LP *p* < 0.05; f = females; m = males).

**Figure 3 ijms-25-03130-f003:**
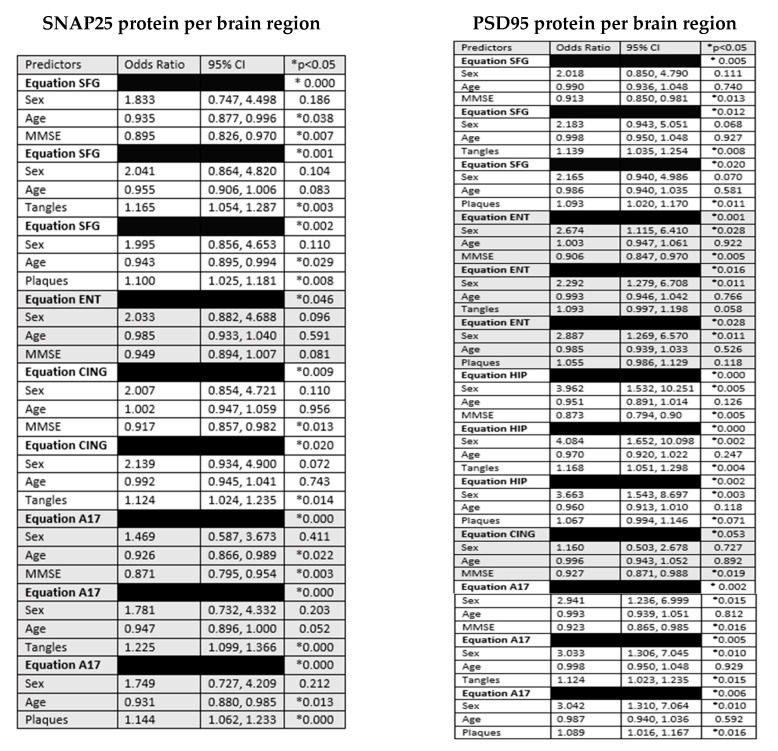
Diagnostic group and sex as logistic regression predictors of regional brain SNAP25 and PSD95 protein concentration. SNAP25 and PSD95 concentrations are the dependent variables. Odds ratios and *p*-values for each independent variable are shown. *p* < 0.05 *; SFG: superior frontal gyrus; ENT: entorhinal cortex; HIP: hippocampus; CING: cingulate; A17: visual cortex area 17.

**Table 1 ijms-25-03130-t001:** General clinical and pathological characteristics of study subjects. Significant group differences were seen for age, brain weight, MMSE, total plaque score, Thal phase, total tangle score, and Braak stage.

DX (*n*)	Age (SD)	Gender (M–F)	PMI Hours (SD)	Brain Weight Grams (SD)	MMSE Out of 30 (SD)	Total Plaques Out of 15 (SD)	Thal Phase 1–5 (SD)	Total Tangle Out of 15 (SD)	Braak Stage I–VI (SD)
CU-LP (33)	83 (7) *	17:16	3 (1)	1211 (123) *	29 (1) *	3 (4) *	1 (2) *	3 (2) *	2 (0.8) *
CU-HP (15)	92 (6) #	7:8	4 (3)	1203 (128)	28 (2)	10 (4) #	3 (1) #	7 (1) #	4 (0) #
MCI (18)	90 (7) #	7:11	4 (2)	1149 (87)	26 (3)	7 (6)	2 (2) #	7 (2) #	4 (0.4) #
ADD (35)	82 (8)	20:15	3 (1)	1089 (125) #	15 (8) #	14 (1) #	5 (.6) #	14 (2) #	6 (0.5) #

* *p* < 0.05 for group comparisons; # *p* < 0.05 for comparison with CU-LP. *n*: number of cases; DX: diagnosis; SD: standard deviations; M: males; F: females; MMSE: Mini Mental State Examination; PMI: postmortem interval in hours; CU-LP: cognitively unimpaired low AD pathology controls; CU-HP: cognitively unimpaired high AD pathology; MCI: mild cognitive impairment; ADD: Alzheimer’s disease dementia.

**Table 2 ijms-25-03130-t002:** Univariable correlations of separate brain region SNAP25 and PSD95 concentrations with MMSE, age, total tangles, and total plaques. Spearman rho (*ρ*) and *p*-values are shown for each correlation. *p* < 0.05 *; *p* < 0.01 **; *p* < 0.001 ***. MMSE: Mini Mental State Examination; A17: primary visual cortex, Brodmann area 17. N/A: non applicable.

SNAP25	MMSE	Age	Total Tangles	Total Plaques	PSD95
Frontal	0.29 **	0.18 (NS)	−0.213 **	−0.292 **	0.655 ***
Entorhinal	0.26 *	0.22 *	−0.050	−0.081	0.737 ***
Hippocampus	0.16 (NS)	0.12 (NS)	−0.085	−0.062	0.698 ***
Cingulate	0.39 ***	0.18 (NS)	−0.294 **	−0.286 **	0.629 ***
A17	0.49 ***	0.32 **	−0.430 ***	−0.445 ***	0.674 ***
**PSD95**	**MMSE**	**Age**	**Total Tangles**	**Total Plaques**	**PSD95**
Frontal	0.31 **	0.06 (NS)	−0.0267 *	−0.213 *	N/A
Entorhinal	0.41 ***	0.20 *	−0.191 *	−0.233 *	N/A
Hippocampus	0.43 ***	0.20 *	−0.324 **	−0.307 **	N/A
Cingulate	0.32 **	0.19 (NS)	−0.257 **	−0.260 **	N/A
A17	0.33 **	0.22 *	−0.231 **	−0.291	N/A

## Data Availability

Data will be available upon reasonable request via email with corresponding author or by visiting www.brainandbodydonationprogram.org.

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
