# Peer review of "Correlation of Presynaptic and Postsynaptic Proteins with Pathology in Alzheimer’s Disease"

_ijms, 2024, doi:10.3390/ijms25063130_

Round 1

Reviewer 1 Report

Comments and Suggestions for Authors

The publication submitted for review investigated the levels of SNAP25 and PSD95 proteins in different brain regions of patients with Alzheimer's disease.

Although the idea of the work seems interesting, the manuscript is written carelessly.

Overall: the text lacks information, the graphs lack bars, the graphs are not clear, the description of the results is wishful thinking, the description of the figures included is missing or mistaken.

In detail:

- There is information in the literature about the SNAP25 and PSD95 proteins, so it is worth disclosing them in the introduction instead the discussion.

- Materials and methods are described inconsistently, some information is in subsections and some is not, information is repeated

- Results: when describing the results it is worth dividing them into subsections, it is worth comparing the text with the graph (you cannot write ‘lower’ as the bar is higher on the graph).

- The statistic is used to show differences. Results that differ significantly should be described.

- Fig.1 one of the results described in the text is missing

- Fig.2 is a table

- Fig.3 is not described (probably the description of this figure is that of Fig.2 - you have to guess)

Author Response

Reviewer #1:

Comments and Suggestions for Authors

The publication submitted for review investigated the levels of SNAP25 and PSD95 proteins in different brain regions of patients with Alzheimer's disease.

Although the idea of the work seems interesting, the manuscript is written carelessly.

Overall: the text lacks information, the graphs lack bars, the graphs are not clear, the description of the results is wishful thinking, the description of the figures included is missing or mistaken.

We thank the reviewer for his/her time reviewing our manuscript. Both graphs had bars representing standard deviations. In order to make figures clearer we added more description on the legend of each figure and tables, as well as the manuscript.

In detail:

- There is information in the literature about the SNAP25 and PSD95 proteins, so it is worth disclosing them in the introduction instead the discussion.

We now added more information on these synaptic proteins in the introduction.

- Materials and methods are described inconsistently, some information is in subsections, and some is not, information is repeated.

We added subsection on methods and added more information as requested by the reviewer.   

- Results: when describing the results, it is worth dividing them into subsections, it is worth comparing the text with the graph (you cannot write ‘lower’ as the bar is higher on the graph).

- The statistic is used to show differences. Results that differ significantly should be described.

Text and results were not conflicting in any figure. Results, table, and graphs were re-written to clarify this reviewer concerns.

- Fig.1 one of the results described in the text is missing

Figure was mentioned fourth row on results, we added additional text to make the description of the results clearer.

- Fig.2 is a table- Correct, we modified this mistake.

- Fig.3 is not described (probably the description of this figure is that of Fig.2 - you have to guess)

Fig2 was mentioned but legend had an error. This was corrected and additional information was added to better describe the figure better.

Reviewer 2 Report

Comments and Suggestions for Authors

The manuscript from Serrano et al. shows changes in synaptic proteins in Alzheimer disease. Results show that synaptic transmission is reduced in this disease because synaptic proteins such as SNAP25 and PSD95 are altered. In addition, SNAP25 and PSD95 concentrations significantly correlated with densities of neurofibrillary tangles and amyloid plaques. They suggest that the reduction of cognition in Alzheimer's patients may be due to this reduction in synaptic transmission. The work is well done and combines different techniques to obtain these conclusions. The methods are well described. However, this work describes these changes in protein concentrations but dii not reach any conclusion. Thus, I have some comments to clarify different issues.

Table 1. Authors indicate the postmortem interval. But, what are the units, hours, days? In what cortical area were these data taken?

Figure 1. The plot shows different scores. In what cortical area were these data taken?

Is there some correlation between cognitive loss and SNAP25 and PSD95 concentrations?

Discussion. There are no conclusions, only description of the data. Therefore, the discussion must be rewritten to explain the changes they have observed.

Are the primary areas more affected than the associative areas?

Why are changes more important in postsynaptic proteins than in presynaptic proteins?

Author Response

Comments and Suggestions for Authors

The manuscript from Serrano et al. shows changes in synaptic proteins in Alzheimer disease. Results show that synaptic transmission is reduced in this disease because synaptic proteins such as SNAP25 and PSD95 are altered. In addition, SNAP25 and PSD95 concentrations significantly correlated with densities of neurofibrillary tangles and amyloid plaques. They suggest that the reduction of cognition in Alzheimer's patients may be due to this reduction in synaptic transmission. The work is well done and combines different techniques to obtain these conclusions. The methods are well described. However, this work describes these changes in protein concentrations but dii not reach any conclusion. Thus, I have some comments to clarify different issues.

Table 1. Authors indicate the postmortem interval. But, what are the units, hours, days? In what cortical area were these data taken?

Figure 1. The plot shows different scores. In what cortical area were these data taken?

We thank the reviewer for their comments and suggestions to make our manuscript better. We now added more information on our manuscript to clarify that our postmortem interval is presented as hours.  We also added more information to clarify that the pathology reported in this study and used for all analysis were summations of all the brain regions we analyzed. “Amyloid plaque (both diffuse and neuritic), and neurofibrillary tangle (NFT) densities were graded blindly as none, sparse, moderate, or frequent using the CERAD templates [36]. All scores were converted to a 0–3 scale for statistical purposes. Regions scored included cortical gray matter from frontal, temporal, parietal, hippocampal CA1, and entorhinal regions, with a maximum summary score of 15. The summation of all scores from all areas were used for all statistical correlations presented in this article.”

Is there some correlation between cognitive loss and SNAP25 and PSD95 concentrations?

Yes, we have univariable and logistic regression showing that MSSE correlated to both proteins expression. We tried to make this clearer now on our manuscript and on the tables.” Univariable correlations were significant, for both SNAP25 and PSD95, with plaque and NFT densities as well as with cognition, specifically MMSE scores, and age in most brain regions (Table 2) but not with brain weight (not shown). Logistic regression models confirmed that the MMSE scores, total plaque scores and total NFT scores correlated significantly with cortical synaptic protein concentrations, were significant even after correcting for sex and age (Table 3).”

Discussion. There are no conclusions, only description of the data. Therefore, the discussion must be rewritten to explain the changes they have observed.

We added more text to make clearer our conclusions. Our main conclusion is that that preventing or reversing synaptic loss might best be accomplished long before the first signs of cognitive decline.

Are the primary areas more affected than the associative areas?

We used association and primary cortices, now added further description in our text to point out we also analyzed visual primary cortices, both brain regions showing high loss of synaptic protein. 

Why are changes more important in postsynaptic proteins than in presynaptic proteins?

Our studies showed changes in both type of proteins, and our results could not be used to imply that one compartment of the synapses is more important than the other. Yet we added to the discussion additional information regarding the complexity of the postsynaptic compartments, and how the myriad of proteins modulating these postsynaptic compartments could become good targets for pharmacological intervention. “Postsynaptic compartments are very complex, constitute of multiple proteins that are constantly being modulated by external and internal signals which make these structure susceptible to pathological changes, but likewise good therapeutic targets.”

Reviewer 3 Report

Comments and Suggestions for Authors

At the manuscript “Correlation of Presynaptic and Postsynaptic Proteins with Pathology in Alzheimer’s Disease” by Dr. Geidy E. Serrano et al authors quantified densities the presynaptic protein SNAP25 and the postsynaptic protein 19 PSD95 in human brain using enzyme-linked immunosorbent assays (ELISA). Protein was extracted from the cingulate, hippocampus, frontal, visual, and entorhinal cortex from cognitively unimpaired controls, subjects with mild cognitive impairment (MCI) and demented subjects with different levels of Alzheimer’s pathology.

 It was shown that SNAP25 is significantly reduced in Alzheimer’s Disease (AD) when compared to controls in frontal cortex, visual cortex and cingulate; while hippocampus showed a smaller, nonsignificant reduction and entorhinal cortex concentrations were not different but all brain areas showed lower PSD95 concentrations in AD when compared to non-demented controls, although in hippocampus this failed to reach significance.

Interestingly, cognitively unimpaired cases with high levels of AD pathology had higher levels of both synaptic proteins in all brain regions. SNAP25 and PSD95 concentrations significantly correlated with densities of neurofibrillary tangles, amyloid plaques and Mini Mental State Examination scores.

Authors believe that obtained results suggest that synaptic transmission is affected by ADD in multiple brain regions. Differences were less marked in entorhinal cortex and hippocampus, most likely due to a ceiling effect imposed by the very early development of neurofibrillary tangles in older people in these brain regions.

The authors conducted a serious study and obtained very interesting results. I have no objections to the essence of the study, but there are some questions.

The authors rightly point out that synaptic integrity is affected by neurodegenerative diseases and correlates with cognitive decline in human and animal models. There are data obtained on animals, which correlate well with the data obtained by the authors.

It would be correct to compare these data authors' data with the citing of the corresponding works, for example:

Plachez et al,  Neuroscience; Amyloid Deposition and Dendritic Complexity of Corticocortical Projection Cells in Five Familial Alzheimer's Disease Mouse; 2023 doi: 10.1016/j.neuroscience.2022.12.013.

This would seriously strengthen the results obtained by the authors.

Quantification synapses in multiple brain regions of the same individuals are needed for both clinical and preclinical studies. Are similar methods available for animal models of AD? If so, can they be used in vivo in long-term, chronic experiments? This is directly related to early diagnosis of AD and the ability to control the progression of the disease.

The presentation of a subject is systematic and comprehensive and analysis is proper. I am happy to recommend the manuscript for the publication after minor corrections mentioned above.

Author Response

Comments and Suggestions for Authors

At the manuscript “Correlation of Presynaptic and Postsynaptic Proteins with Pathology in Alzheimer’s Disease” by Dr. Geidy E. Serrano et al authors quantified densities the presynaptic protein SNAP25 and the postsynaptic protein 19 PSD95 in human brain using enzyme-linked immunosorbent assays (ELISA). Protein was extracted from the cingulate, hippocampus, frontal, visual, and entorhinal cortex from cognitively unimpaired controls, subjects with mild cognitive impairment (MCI) and demented subjects with different levels of Alzheimer’s pathology.

 It was shown that SNAP25 is significantly reduced in Alzheimer’s Disease (AD) when compared to controls in frontal cortex, visual cortex and cingulate; while hippocampus showed a smaller, nonsignificant reduction and entorhinal cortex concentrations were not different but all brain areas showed lower PSD95 concentrations in AD when compared to non-demented controls, although in hippocampus this failed to reach significance.

 Interestingly, cognitively unimpaired cases with high levels of AD pathology had higher levels of both synaptic proteins in all brain regions. SNAP25 and PSD95 concentrations significantly correlated with densities of neurofibrillary tangles, amyloid plaques and Mini Mental State Examination scores.

 Authors believe that obtained results suggest that synaptic transmission is affected by ADD in multiple brain regions. Differences were less marked in entorhinal cortex and hippocampus, most likely due to a ceiling effect imposed by the very early development of neurofibrillary tangles in older people in these brain regions.

 The authors conducted a serious study and obtained very interesting results. I have no objections to the essence of the study, but there are some questions.

 The authors rightly point out that synaptic integrity is affected by neurodegenerative diseases and correlates with cognitive decline in human and animal models. There are data obtained on animals, which correlate well with the data obtained by the authors.

 It would be correct to compare these data authors' data with the citing of the corresponding works, for example:

Plachez et al,  Neuroscience; Amyloid Deposition and Dendritic Complexity of Corticocortical Projection Cells in Five Familial Alzheimer's Disease Mouse; 2023 doi: 10.1016/j.neuroscience.2022.12.013.

This would seriously strengthen the results obtained by the authors.

We want to thank the reviewer for the thorough review and summary of our study. We added now Plachez manuscript and added more discussion comparing our results with other studies. “Total densities of plaques and tangles also correlated with the concentrations of synaptic protein, but even though we found statistically significant correlations, the correlations were not as strong as expected, suggesting other underlying mechanisms that should be further explored. Similar results were recently reported by Plachez, et al 2023 were they showed that synaptic modulation in an AD mouse model was observed in primary and association cortices independently of amyloid accumulation.”

 Quantification synapses in multiple brain regions of the same individuals are needed for both clinical and preclinical studies. Are similar methods available for animal models of AD? If so, can they be used in vivo in long-term, chronic experiments? This is directly related to early diagnosis of AD and the ability to control the progression of the disease.

The ELISAs describes in this study could be used in brains from animal models and in biofluids such as CSF. In the study we emphasize that now we have synaptic PET ligands that could allow the community to explore further synaptic changes during life and aging. It will be important to understand how those change as we age normally or with a neurodegenerative disorder such as ADD.  

 The presentation of a subject is systematic and comprehensive and analysis is proper. I am happy to recommend the manuscript for the publication after minor corrections mentioned above.

We thank again the reviewer for the support and sponsorship for publishing our manuscript.

Round 2

Reviewer 1 Report

Comments and Suggestions for Authors

The introduction of the manuscript has been corrected. Nevertheless, there are still discrepancies between the description of the results and Fig 2.

Besides, the results obtained for the expression of SNAP25 and PSD95 proteins require confirmation by another method, such as western-blot - at least in selected groups and selected structures.

Author Response

(The authors gave the same response as above.)

Reviewer 2 Report

Comments and Suggestions for Authors

The authors appropriately addressed all concerns raised with the initial manuscript submission

I have no more suggestions to make.

Author Response

(The authors gave the same response as above.)
